# Effect of High-Intensity Interval Training Combined with L-Citrulline Supplementation on Functional Capacities and Muscle Function in Dynapenic-Obese Older Adults

**DOI:** 10.3390/jcm7120561

**Published:** 2018-12-17

**Authors:** Fanny Buckinx, Gilles Gouspillou, Livia P. Carvalho, Vincent Marcangeli, Guy El Hajj Boutros, Maude Dulac, Philippe Noirez, José A. Morais, Pierette Gaudreau, Mylène Aubertin-Leheudre

**Affiliations:** 1Département des Sciences de l’activité physique, Groupe de Recherche en Activité Physique Adaptée (GRAPA), Université du Québec à Montréal, Montréal, QC H2X 1Y4, Canada; fanny.buckinx@uliege.be (F.B.); gilles.gouspillou@gmail.com (G.G.); livia.pinheiro.carvalho@gmail.com (L.P.C.); vincentmarcangeli@gmail.com (V.M.); guyhajj38@hotmail.com (G.E.H.B.); philippe.noirez@parisdescartes.fr (P.N.); 2Centre de Recherche de l’Institut Universitaire de Gériatrie de Montréal (CRIUGM), Montréal, QC H3W 1W6, Canada; dulacmaude.uqam@gmail.com; 3Institut de Recherche bioMédicale et d’Épidémiologie du Sport (IRMES), Université Paris Descartes, Sorbonne Paris Cité, 75012 Paris, France; 4Division of Geriatric Medicine, McGill University Health Centre (MUHC), McGill University, Montréal, QC H3A 1A1, Canada; jose.morais@mcgill.ca; 5Département de Médecine de l’Université de Montréal, Centre de Recherche du Centre Hospitalier Universitaire de Montréal (CRCHUM), Université de Montréal, Montréal, QC H2X 3E4, Canada; pierrette.gaudreau@umontreal.ca

**Keywords:** HIIT, citrulline, dynapenia, obesity, functional capacities, muscle function, aging

## Abstract

Background: To compare the effects of high-intensity interval training (HIIT) alone vs. HIIT combined with L-citrulline (CIT) supplementation on functional capacity and muscle function in dynapenic-obese elderly. Methods: A total of 56 obese (fat mass: men > 25%, women > 35%) and dynapenic (grip strength/body weight: women < 0.44, men < 0.61) subjects were recruited and divided in two groups: HIIT+CIT (*n* = 26; age: 6 5 ± four years) vs. HIIT+Placebo (PLA, *n* = 30; age: 68 ± four years). Participants followed a 12-week HIIT using an elliptical trainer. Participants took a single and isocaloric 10 g-dose of CIT or PLA every day. Body composition; functional and aerobic capacities; absolute or relative upper and lower limbs muscle strength, muscle power; and energy balance were measured pre and post intervention. Results: Both groups significantly improved functional capacity and muscle function. However, HIIT+CIT demonstrated greater improvements in fast-paced Timed Up & Go (*p* = 0.04) and upper limbs muscle strength (absolute and relative) (*p* = 0.05) than HIIT+Placebo. Conclusion: CIT supplementation when combined with HIIT seems to induce greater improvements in upper limbs muscle strength and walking speed in dynapenic-obese elderly. Further studies are needed to confirm our results, to elucidate the mechanisms underlying the beneficial effects of CIT and to define the long-term impact of CIT/HIIT.

## 1. Introduction

Aging is often associated with a progressive loss of muscle strength (dynapenia) and an increase in fat mass (obesity), both leading to physical declines [1]. The coexistence of dynapenia and obesity, affecting 7.6 to 11.1% of the population [2], is associated with a greater decline in functional capacities than obesity or dynapenia alone [3,4,5]. Developing novel strategies to counteract dynapenic-obesity is therefore of major relevance to promote healthy aging [6].

Exercise interventions have been shown to improve multiple parameters related to dynapenia or obesity, including muscle function, fat mass, as well as functional and aerobic capacities. However, and despite this knowledge, almost 60% of older adults are sedentary [7]. One of the main reported barriers to engaging in physical activity is the lack of time [8]. Interestingly, it has been shown that cardiovascular training is as beneficial as resistance training to maintain muscle quality and to mitigate the decline in functional capacities in individuals over the age of 60 [9]. High-intensity interval training (HIIT), a subtype of cardiovascular training has been shown to be particularly effective in triggering beneficial adaptations. Indeed, among diabetic older people, a 12-week HIIT has been shown to improve body composition (i.e., reduction in waist-to-hip circumference and appendicular fat and increase in lean mass) and cardiorespiratory fitness (maximal oxygen consumption; VO_2max_) [10]. A recent meta-analysis has demonstrated that HIIT can significantly increase VO_2max_ while reducing cardio-metabolic risk factors (i.e., waist circumference, body fat, diastolic blood pressure, and fasting glucose levels) in overweight adults [11]. Evidence also indicates that high-intensity exercise, even when performed during short periods of time, can induce greater improvements in functional capacities, body composition, and aerobic capacity vs. moderate-intensity exercise in the elderly [12,13,14]. Knowles et al. showed that HIIT resulted in significant increase in aerobic capacity in both sedentary and active older adults [15]. Sculthorpe et al. also reported in old individuals a significant increase in muscle mass and power and a significant decrease in total body fat following HIIT [16]. Although one might question how well HIIT can be tolerated by older adults, it is important to highlight here that previous studies have shown that HIIT elicited similar or even higher enjoyment and adherence levels than moderate-intensity continuous training [17,18]. 

Nutritional strategies also have the potential to improve body composition in dynapenic or obese older adults [6]. Amongst these strategies, the supplementation with the non-proteinogenic amino acid called L-citrulline (CIT) [19], an intermediate of the urea cycle produced in the liver from arginine during nitric oxide production [20], is of particular interest. First, and in contrast with arginine or other nutrients such as omega-3 or proteins, CIT escapes splanchnic extraction [19]. It has also been suggested that CIT supplementation can influence muscle and fat metabolism in animal and human models. Indeed, in aged rats, CIT supplementation seems to protect against oxidative damage to circulating lipids and lipoproteins [21] and to induce lipolytic and anti-glyceroneogenic effects via stimulation of fatty acid release, ultimately leading to a reduction in adipose tissue mass [22]. Faure et al. [23] found that muscle mass and maximal tetanic isometric strength significantly increased while refeeding aged malnourished rats with CIT or Leucine pulse. In old rats, CIT supplementation (1 g/kg/day) increased muscle mass, muscle fiber size, and the expression and activity of mitochondrial proteins [21]. Recently, Bouillanne et al. have also reported an increase in lean mass and a decrease in fat mass following CIT supplementation in malnourished older women [24]. 

Nutrition in combination with exercise is considered optimal for maintaining muscle function. Figueroa et al. recently showed that vibration/strength training combined with CIT induces greater improvement in leg fat-free mass in obese post-menopausal women [25]. However, no study has investigated the potential effects of CIT supplementation combined with an exercise intervention in older men and women, especially in individuals at high risk of physical disability. Because both CIT and HIIT exert positive effects on muscle function and body composition, combining these two interventions might represent an effective strategy to improve health in dynapenic-obese older adults. To test this hypothesis, the present study compared the effects of HIIT alone vs HIIT combined with CIT supplementation on functional capacities and muscle function in dynapenic-obese older adults.

## 2. Material and Methods

### 2.1. Study Design

This is a secondary-analysis from a double blind randomized trial. The randomization was performed by blocks of four by computer-generated randomization procedure.

All procedures were approved by the Ethics Committee of the “Université du Québec à Montréal (UQAM)”. All participants provided informed written consent after having received information on the nature, goal, procedures and risks associated with the study.

### 2.2. Population

Subjects were recruited from the community via social communication (flyers and meetings in community centers) in the Great Montreal area. All participants followed an exercise intervention (HIIT) and received an isocaloric supplementation (CIT or placebo). Thus, participants were randomly and double blindly assigned to HIIT+CIT or HIIT+PLA groups.

To be included in the main study, subjects had to meet the following criteria: (1) aged 60 years and over, (2) inactive for at least 6 months (<2 h/week of structured exercise), (3) stable weight (±2 kg) over the past 6 months, (4) no orthopedic limitations, (5) no counter-indication to practice physical activity (assessed using the Physical Activity Readiness Questionnaire), (6) absence of menstruation for the past 12 months for women, (7) non smoker, (8) low alcohol consumption (≤2 drinks/day). Subjects with diagnosed (untreated) neurological, cardiovascular, lung diseases or cognitive disorders were also excluded.

To be included in this secondary-analysis study, subjects had to also meet the following specific criteria: (1) dynapenic (upper limb muscle strength (kg)/body weight (kg) < 0.44 (women) and <0.61 (men) [26], and (2) obese (fat mass: Men > 25%, Women > 35%; [27]).

### 2.3. Intervention

#### 2.3.1. Exercise Training

Participants followed a high-intensity interval training (HIIT) on an elliptical device, which was chosen to reduce impacts on lower extremity joints [10]. The intervention was performed three times per week in non-consecutive days during 12 weeks and was supervised by trained health professionals. The intensity of each cycle was based on percentage of maximal heart rate and/or perceived exertion (Borg scale) [28] or exclusively based on the later in case of anti-arhythmic and inotropic agents use. The maximal heart rate was determined using the validated equation of Karvonen [((220-age) – Heart Rate rest) × % Heart Rate target] + Heart Rate rest [29]. More specifically, the 30 min exercise session consisted of a 5 min warm-up at a low intensity (50–60% maximal heart rate and/or a score between 8 and 12 on the Borg scale); a 20-min HIIT of multiples 30 s sprints at a high intensity (80–85% maximal heart rate or Borg’ scale >1 7) alternating with sprints of 90 s at a moderate intensity (65% maximal heart rate or Borg’scale score 13–16); and a 5 min cool-down (50–60% maximal heart rate and/or a Borg’ scale score 8–12). To ensure that maximal heart rate was always above 80% during high-intensity intervals, speed and resistance of the elliptical device were continuously adjusted throughout the training session. Participants needed to complete 80% or more of their training sessions in order to be included in the analysis [10]. 

#### 2.3.2. L-citrulline Supplementation

During the 12 weeks, participants in the HIIT+CIT group took a single daily dose of 10 grams of L-CIT (Citrage ©) containing 38 kcal per dose, while participants in the HIIT+PLA group took a single dose of a placebo powder (maltodextrin) equivalent in weight, appearance, taste and calories (see Table 1). Supplements were taken every day during lunch meal. The dose of CIT ingestion was based on the results of Moinard et al. [21]. An independent technician prepared and identified each individual’s container containing PLA or CIT with a letter code. Each container had a sealed envelope containing the nature of the supplement that was securely stored in a locked container. Evaluators and experimentators were kept blind until the end of the protocol. Data on supplementation intake compliance was monitored by giving one box of the supplementation per month and by randomly assigning different quantities of powder packets per box to participants. 

### 2.4. Measurements

Socio-demographic characteristics (age and sex), cognitive status [30], body composition, aerobic and functional capacities at baseline and after 12 weeks (i.e., at the end of the intervention) were assessed for each participant in the same order.

The validated Montreal Cognitive Assessment (MoCA) was used to assess cognitive status. The MoCA test evaluates visuospatial abilities, executive functions, orientation, language, attention, concentration, and working memory. An extra point to the total score is given if the subject has ≤12 years of education. A cut-off of ≥26 out of 30 points was established to differentiate healthy individuals from individuals with mild cognitive impairment [30].

#### 2.4.1. Body Composition

Body weight and height were determined using an electronic scale (Omron HBF-500CAN) and a stadiometer (Seca), from which Body Mass Index (BMI = Body mass (Kg)/Height (m^2^)) was calculated. Waist circumference was measured to the nearest 0.1 cm.

Dual-energy X-ray absoptiometry [DXA] (GE Prodigy Lunar) was used to assess fat (total, android, gynoid, and legs, in %) and lean (total, in kg) masses. Coefficients of variations for fat mass and lean body mass measures were previously validated in ten young adults (measured one week apart) were 5.7 and 1.1%, respectively [31]. 

#### 2.4.2. Functional and Aerobic Capacities

Five different validated tests were performed to assess functional capacities: 

Walking speed was estimated using the “Timed Up & Go” test (in seconds). This test, which consists in standing from a chair, walking a 3 m distance and sitting down again [32], was performed at a comfortable and self-paced and at a fast-paced walking speed. A duration above 30 s indicates limited mobility and an increased risk of falling whereas a duration of less than 20 s indicates appropriate mobility with subject likely to be independent in activities of daily living [33]. 

Balance was assessed using the validated unipodal balance test. Participants were standing on both legs and alternately standing on the right and left leg with eyes opened and arms by the side of the trunk. The time was recorded in seconds from the moment one foot was lifted from the floor to the moment when it touched the ground, the stance leg moved, or until 60 s had elapsed [34]. 

Lower-body function was measured using the chair stand test. Subjects were asked to stand up from a sitting position and to sit down 10 times as fast as possible, with arms across their chests [35]. The time (in seconds) to perform the task was recorded. 

Weight shifting ability in the forward and upward directions was estimated using the alternate-step test. Participants were placed facing toward a 20 cm height step and instructed to touch its top with the right and left foot, alternatively, as fast as possible during a 20 s period [36,37]. The number of steps was recorded for analysis. 

Mobility and aerobic capacities were evaluate using the 6 min walking test. Participants were asked to walk as much as possible during 6 min. In each minute of the test, volunteers received the same standardized encouragement according to the ATS/American College of Chest Physicians recommendations for the six-minute walking test [31]. Participants were allowed to interrupt and return to exercising as well as to reduce or increase speed according to perceived effort [38]. The distance, in meters, was recorded and used to estimate aerobic capacity (maximal oxygen consumption; VO_2max_) according to the following equation: VO_2max_ (mL·k^−1^·min^−1^): 70.161 + (0.023 × distance [m]) − (0.276 × body weight [kg]) − (6.79 × sex [Men = 0, Women = 1]) − (0.193 × heart rate [pulse/min]) − (0.191 × age [years]) [39]. 

#### 2.4.3. Muscle Function

Maximum voluntary upper limb muscle strength was measured using a hand dynamometer with adjustable grip (Lafayette Instrument). This method is reliable [40]. Participants were standing upright with the arm along the side of the body with the elbow extended and the palm of the hand facing the thigh. Participants were advised to squeeze as hard as possible the hand dynamometer for up to 4 s. Three measurements for each hand, alternatively, were performed and the maximal score for each was recorded. Upper limb muscle strength was expressed in absolute (kg) and relative (divided by body weight (BW; kg/kg)) values [41]. 

Dynapenic status was assessed as previously described [26], with type I dynapenia corresponding to a value ranging from 1 to 2 standard deviations (SDs; women: 0.44 to 0.35 kg/BW and men: 0.61 to 0.50 kg/BW) and type II dynapenia corresponds to a value 2 SDs below (women: <0.35 kg/BW and men: <0.50 kg/BW) the mean value of our young reference population [26]. This reference population was composed of 15 young healthy women and 15 young healthy men aged 18 to 30 years. During their visit, body composition, maximum isometric strength of the knee extensors, and handgrip strength were measured to be able to define dynapenia indexes. The use of this dynapenia index (handgrip strength/body weight) is based on the work of Barbat-Artigas and colleagues who showed using a wide battery of clinical markers that this dynapenia index is the best predictor of functional incapacities [42].

Maximal isometric lower limb muscle strength was assessed using a strain gauge system attached to a chair (Primus RS Chair, BTE) upon which participants were seated with the knee and hip joint angles set at 135° and 90°, respectively. The knee angle was set to 135°, compared to the typical 90°, in order to diminish the maximal joint torque that could be generated [43,44], particularly in light of generally more fragile bones in the elderly [45]. The tested leg was fixed to the lever arm at the level of the lateral malleoli on an analog strain gauge to measure strength. The highest of three maximum voluntary contractions was recorded [42]. Lower limb muscle strength was expressed in absolute terms (N) and relative to body weight (divided by body weight (kg/kg)). 

Upper and lower limb muscle quality was calculated using the maximal grip strength (kg) divided by arm lean mass (kg; DXA) and the maximal knee extensor strength (in kg) divided by leg lean mass (kg; DXA), respectively [46]. Both of these indices are known to be related with functional impairments [42,47]. 

Lower limb muscle power was measured using the Nottingham Leg Extensor Power rig [48] with participants in a sitting position. Participants were asked to push the pedal down as hard and fast as possible, accelerating a flywheel attached to an A–D converter [49]. Power was recorded for each push until a plateau/decrease was observed. This assessment has been demonstrated to be safe, sensitive, and reliable in older adults [49].

#### 2.4.4. Energy Balance

As previously described and validated in the elderly population, dietary intake was assessed before and after the intervention using the 3-day food record method (two weekdays and one weekend day) [50]. Participants were asked keep their dietary habits during the intervention period. Analyzes of total intake as well as protein, lipids, or carbohydrate intake were performed using the software Nutrific© according to the standardized Canadian Food file (CNF2015).

The number of steps were used to estimate the level of physical activity using a validated tri-axial accelerometer (SenseWear® Mini Armband) as previously described [51,52]. Participants had to wear the device on the left arm all the time during 7 consecutive days, except when taking a shower or swimming. Each participant had to wear the device at least 85% of the time to be included in the study.

### 2.5. Statistical Analysis

Data distributions were tested using the Kolmogorov test. Quantitative variables were expressed as mean ± standard deviation (SD). Qualitative variables were expressed as percentage. An independent sample t-test was used to identify between-group baseline differences. The Chi-squared test or Fisher test were used to compare frequency of observations between groups and the prevalence of dynapenia. A two-way repeated-measure analysis of covariance (ANCOVA adjusted for age) was used to estimate time (HIIT intervention) and time*group (PLA vs. CIT group) effects. Individuals improving their physical parameters were identified as responders; individuals displaying no change or a decrease in their physical paramters were identified as non-responders. All statistical analyses were performed using SPSS 25.0 (Chicago, IL, USA) and Statistica 10. *p* ≤ 0.05 was considered statistically significant.

## 3. Results

### Population

A total of 107 subjects were recruited through advertisements in a local newspaper and Montreal communities. Among them, 98 accepted to take part in the main study and were randomly and double blindly divided into two groups: HIIT+CIT and HIIT+PLA groups. By design, out of these 98 subjects, 56 older adults were considered obese and dynapenic and completed the intervention: HIIT+CIT (*n* = 26) vs. HIIT+PLA (*n* = 30) (Figure 1).

Main descriptive characteristics of the population (age, sex, cognitive status, body composition) at baseline and after 12 weeks of intervention are presented in Table 2. With the exception of age (*p* = 0.03), all baseline characteristics were comparable between groups. Subjects in HIIT+PLA group were significantly older than those in the HIIT+CIT group (68 ± 4 vs. 66 ± 4 years respectively). All subsequent statistical analyzes were therefore adjusted for age. Nevertheless, it is important to note that with or without adjustment for age our results and conclusions are identical. 

Our results demonstrate a decrease in waist circumference in both groups after 12 months of intervention. Moreover, HIIT+CIT group significantly decreased total, android and legs fat masses (*p* ≤ 0.05). HIIT+PLA group displayed an increase in total lean mass (*p* ≤ 0.05). No within or between group effects were observed for body composition (Table 2). 

Concerning functional capacities and muscle function (Table 3, Figure 2), we observed an improvement for all of these variables (*p* ≤ 0.05) following the intervention period in HIIT+CIT group. Almost the same observations were made in the group HIIT+PLA, excepted for upper limb muscle strength, upper muscle quality, lower limb muscle strength, and lower muscle quality which were not significantly altered at the end of the intervention. A significant within group interaction was observed for chair test (*p* = 0.03), normal-paced (*p* = 0.03) and fast-paced (*p* = 0.04) walking speed, upper limb muscle strength (*p* = 0.035) and relative upper limb muscle strength (*p* = 0.019). Finally, HIIT+CIT displayed greater improvements in fast-paced “Timed Up and Go” test (*p* = 0.04), upper limb muscle strength (*p* = 0.05) and relative upper limb muscle strength (*p* = 0.05). HIIT+CIT also displayed a trend for greater improvement in lower limb muscle strength (*p* = 0.07) than the HIIT+PLA group (Table 3). The evolution of these three variables, in both groups, during the intervention is shown in Figure 2.

As shown in Table 4, the intervention did not affect energy balance in both groups. Furthermore, both groups kept the same lifestyle habits throughout the intervention since no within or between group effects were highlighted. 

The percentage of responders ranged from 53.8% (lean mass and gynoid fat masses) to 100% (fast-paced “Timed Up and go” and chair stand tests, alternate step test) in the HIIT+CIT group and from 46.7% (android fat mass) to 100% (alternate step test) in the HIIT+PLA group. Overall, the rate of responders is very high for all functional capacities (>80%) in both groups (Table 5).

It is also important to note that 12% of our subjects became non dynapenic in the HIIT+CIT group after the intervention period. In the HIIT+PLA group, none of the dynapenic subject at baseline became non-dynapenic at the end of the study (between group difference: *p* = 0.88). 

## 4. Discussion

The present study indicates that HIIT is an efficient intervention to increase muscle function and functional capacities in dynapenic-obese older adults and that adding CIT supplementation results in further improvements in walking speed and upper muscle strength. First, independently of supplementation with CIT, HIIT results in significant improvements in functional and aerobic capacities, muscle function and body composition. Our results are consistent with recent meta-analyses in adults with obesity suggesting that HIIT is effective to improve VO_2max_ and cardiometabolic risk factors (waist circumference, body fat, diastolic blood pressure, and fasting glucose levels) [11,53]. In addition, our results strengthen the available literature showing that exercise interventions are effective in improving physical capacity (such as balance and stair tests) in dynapenic-obese older adults as suggested by Senechal et al. [54] and that HIIT may also improve muscle function and functional capacities in older adults [16,55]. However, existing studies [11,16,53,55] are heterogeneous in the studied populations, training modalities, follow-up durations, and measured outcomes which hampers comparisons with our findings. Indeed, HIIT embraces a variety of interval protocols with varying duration and interspersed recovery breaks [56]. Under our conditions (3 times/weeks; 30 min/session on elliptical), HIIT is feasible, with a 90% adherence rate, even in older adults. To the best of our knowledge, this is the first study to demonstrate the beneficial physical and functional impact of HIIT in dynapenic-obese older. 

Importantly, our results indicate that combining HIIT and CIT results in greater improvements in upper muscle strength and walking speed vs. HIIT-PLA in dynapenic-obese older adults. Few studies have investigated the effects on CIT or exercise alone on muscle function (muscle mass, strength, physical performance, functional capacity or mobility) in elderly population. Our results strengthen the view that CIT supplementation can improve functional and muscular capacities in Humans [19]. In contrast with Bouillanne et al. who reported data indicating greater beneficial effects of CIT on appendicular skeletal muscle mass and fat mass in malnourished older women [24], we did not observe additional benefit of CIT on muscle and fat masses. These discrepancies could be explained by differences in nutritional status of our population (living in the community and well-nourished here vs. malnourished in [24]), the study design (with exercise intervention here and without exercise intervention in [24]) or the duration (12 weeks here vs. 3 weeks in [24]). In addition, it is possible that HIIT, due to its effectiveness, might have masked potential positive impacts of CIT on muscle and adipose tissues [57]. To date, only one study has investigated the effects of CIT supplementation combined with a whole-body vibration training in postmenopausal women and concluded that the group receiving CIT had a greater improvement on leg muscle function than those not receiving any supplementation [25]. Although caution should be taken in comparing our results from those obtained by Figueroa et al. [25] due to differences in study populations (post-menopausal women in [25] versus older adults here) and type of exercise/supplementation, our conclusion is in line with those from Figueroa et al. since we also observed greater improvements in muscle function in HIIT+CIT vs. HIIT+PLA (upper limb muscle strength, relative upper limb muscle strength and fast-pace “Timed Up and Go” test). 

Based on available literature [24] and our own findings, one could hypothesize that insufficient arginine availability might not be restricted to malnourished older adults but might also be present in all older adults. Importantly, arginine levels in dietary intake is not the only parameter to consider since a significant portion of the ingested arginine is metabolized both in gut and liver for their own need. Because CIT escapes splanchnic extraction and based on the available literature showing that CIT effects on adipose tissue explants in rats are mediated by nitric oxide (NO) [21], it is possible that CIT could restore arginine availability to adequate level and mediates its positive effects through NO in older adults.

In addition, the present study could have important clinical implications for non-pharmacological prescription in older populations with the coexistence of dynapenia and obesity, a combination resulting in greater functional decline than either obesity or dynapenia alone. Interestingly, HIIT combined with CIT supplementation could lead to beneficial effects on physical and muscle function in this specific population as evidenced by our findings. It is also important to highlight that HIIT+CIT decreased the prevalence of dynapenia in our study.

Overall, our results reinforce the need to further investigate the potential effect of CIT supplementation on functional and mobility decline in older adults. Despite these novel findings, some limitations must be addressed. First, our results are limited by design since this study is a secondary-analysis from a larger clinical trial with a relatively small sample size. Although sample size has been initially calculated for the primary outcome of interest in our study (walking speed), our study may have been underpowered for detecting between-group significant effects on some secondary outcomes. In addition, due to small sample sizes, sex-specific analyzes could not be performed. Even if appropriate corrections were performed here, it remains that HIIT+PLA was significantly older than HIIT+CIT. It is also important to mention here that our sample was only composed of dynapenic-obese older adults and thus, our findings are limited to this specific population. The absence of a group receiving only CIT supplementation is another potential limitation of this study. It is worth noting that a wide range of assessments using validated techniques were used in the present study. Finally, another limitation is that the study provides no mechanistic insight into the mechanisms (i.e., mitochondrial activity, GH axis, nitric oxide etc.) underlying the beneficial impact of CIT.

## 5. Conclusions

The present study highlights that CIT supplementation combined with HIIT is feasible and effective to improve overall walking speed and upper muscle strength in dynapenic-obese older adults. These findings might have important implications for public health since they indicate that HIIT+CIT represent an attractive strategy to delay or prevent loss of autonomy in dynapenic-obese older adults. Such a statement is supported by the decreased prevalence of dynapenia induced by this combined non-pharmacological strategy. Further studies are now required to define the mechanisms underlying the protective effects of HIIT+CIT as well as the long term impact of this intervention. 

## Figures and Tables

**Figure 1 jcm-07-00561-f001:**
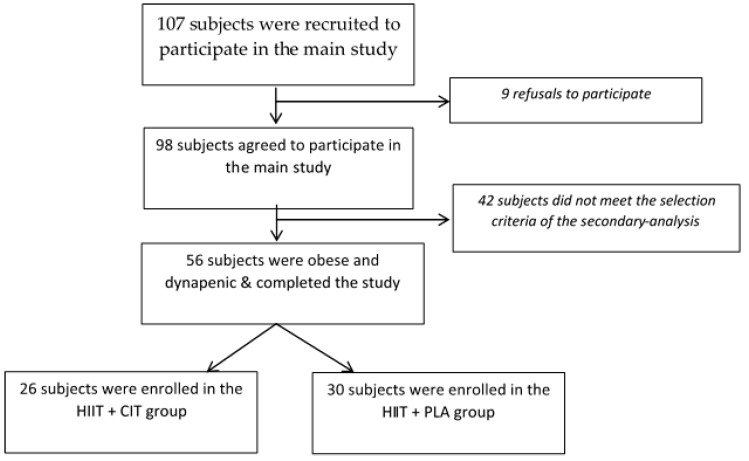
Flow chart of the Secondary-ANALYSIS study.

**Figure 2 jcm-07-00561-f002:**
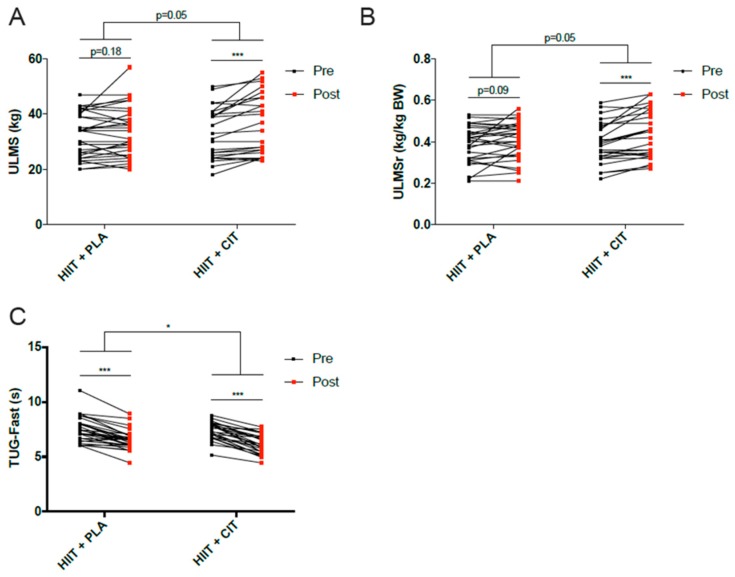
Evolution of ULMS (**A**), ULMSr (**B**) and TUGF (**C**) during the intervention in both groups. Significant differences between HIIT-CIT and HIIT-PLA groups at baseline using *t* test significant differences Time (HIIT intervention) and time × group effects using repeated-measure ANCOVA (adjusted by age). *: *p* < 0.05; ***: *p* < 0.0001. TUGF = fast-paced Timed up and Go; ULMS = Upper Limb Muscle Strength; ULMSr = Upper Limb Muscle Strength/ body weight.

**Table 1 jcm-07-00561-t001:** Citrulline and placebo nutrients composition.

Citrulline (10 g)	Placebo (10 g)
38 kcal	38 kcal
0 g of proteins	0 g of proteins
0 g of maltodextrin (carbohydrate)	10 g of maltodextrin (carbohydrate)
10 g of L-citrulline	0 g of L-Citrulline

**Table 2 jcm-07-00561-t002:** Characteristics and differences in body composition at baseline and after 12 weeks of intervention between groups.

Variables	HIIT-CIT (*n* = 26)			HIIT-PLA (*n* = 30)			*p*-Value	
	Pre	Post	*p* Value *	Pre	Post	*p* Value *	Time Effect Adjusted for Age	Time × Group Effect Adjusted for Age
General characteristics								
Age (years)	65.7 ± 4.2 *^a^*	NA	NA	68.1 ± 4.2 *^a^*	NA	NA	NA	NA
Sex (% men)	50	NA	NA	50	NA	NA	NA	NA
MoCA (/30)	27.1 ± 1.9	NA	NA	27.9 ± 1.4	NA	NA	NA	NA
Body composition								
Body weight (BW; kg)	82.6 ± 12.5	82.1 ± 10.9	0.40	83.7 ± 11.8	82.6 ± 12.5	0.30	0.35	0.98
BMI (kg/m²)	30.5 ± 4.1	30.4 ± 4.6	0.56	30.5 ± 4.9	30.0 ± 4.9	0.27	0.67	0.62
WC (cm)	107 ± 11	104 ± 11	<0.001	106 ± 11	104 ± 10	<0.001	0.20	0.15
Total LM (kg)	47.8 ± 7.4	48.2 ± 7.3	0.19	47.3 ± 9.2	47.9 ± 9.6	0.04	0.54	0.59
Total FM (%)	38.9 ± 5.8	37.8 ± 6.3	0.02	38.9 ± 6.3	38.5 ± 7.6	0.25	0.23	0.17
Android FM (%)	48.8 ± 5.5	47.7± 6.8	0.09	48.5 ± 6.6	48.1 ± 7.2	0.46	0.41	0.25
Gynoid FM (%)	40.4 ± 8.1	39.6 ± 8.8	0.19	41.1 ± 10.1	40.8 ± 10.2	0.55	0.42	0.46
Leg FM (%)	36.5 ± 9.0	35.5± 9.2	0.02	37.0 ± 10.7	36.5 ± 10.8	0.18	0.17	0.40

Data are presented as means ± SD. *p* ≤ 0.05: significant. NA: not applicable. Significant differences between HIIT-CIT and HIIT-PLA groups at baseline using *t* test significant differences. * significant intra-group differences between pre and post intervention using paired t test. Time (HIIT intervention) and time × group effects using repeated-measure ANCOVA (adjusted by age). BW = Body Weight; BMI = Body Mass Index; WC = waist circumference; LM = Lean Mass; FM = Fat Mass

**Table 3 jcm-07-00561-t003:** Characteristics and differences on muscle function and functional capacities at baseline and after 12 weeks of intervention between groups.

Variables	HIIT-CIT (*n* = 26)			HIIT-PLA (*n* = 30)			*p*-Values	
	Pre	Post	*p* Value *	Pre	Post	*p* Value *	Time Effect Adjusted for Age	Time × Group Effect Adjusted for Age
Functional & aerobic capacities								
TUGn (s)	9.9 ± 1.3	8.7 ± 0.9	<0.001	10.3 ± 1.8	9.1 ± 1.3	<0.001	0.03	0.53
TUGf (s)	7.4 ± 0.8	6.2 ± 0.9	<0.001	7.5 ± 1.1	6.6 ± 0.9	<0.001	0.04	0.04
6 MWT (m)	558 ± 92	633 ± 85	<0.001	550 ± 85	618 ± 91	<0.001	0.61	0.70
Estimated VO_2max_ (mL/min/kg)	17.8 ± 2.1	19.5 ± 1.9	<0.001	17.6 ± 2.1	19.2 ± 2.1	<0.001	0.67	0.69
Unipodal balance (/60 s)	26.7 ± 18.6	40.5 ± 21.6	0.001	22.4 ± 14.6	34.5 ± 20.2	<0.001	0.62	0.48
Chair stand test (s)	19.1 ± 3.3	15.1 ± 2.7	<0.001	18.8 ± 3.7	15.6 ± 3.7	<0.001	0.03	0.15
Alternate step test (*n*)	30.3 ± 4.9	35.0 ± 5.6	<0.001	28.9 ± 3.9	33.6 ± 4.7	<0.001	0.59	0.56
Muscle Function								
ULMS (kg)	32.6 ± 9.1	35.7 ± 10.8	<0.001	32.4 ± 8.1	33.5 ± 9.2	0.18	0.035	0.05
ULMSr (Kg/Kg)	0.39 ± 0.09	0.43 ± 0.11	<0.001	0.39 ± 0.08	0.41 ± 0.08	0.09	0.019	0.05
Upper MQ (kg/kg)	6.20 ± 1.04	6.79 ± 1.19	0.004	6.11 ± 1.23	6.98 ± 4.50	0.21	0.19	0.98
LLMS (*N*)	348 ± 83	379 ± 72	0.007	339 ± 92	347 ± 96	0.22	0.07	0.07
LLMSr (kg/kg)	0.42 ± 0.09	0.46 ± 0.08	<0.001	0.42 ± 0.11	0.43 ± 0.09	<0.001	0.10	0.14
Lower MQ (kg/kg)	2.11 ± 0.45	2.26 ± 0.35	0.004	2.06 ± 0.35	2.08 ± 0.31	0.15	0.11	0.10
Muscle Power (W)	153 ± 52	186 ± 56	<0.001	155 ± 70	186 ± 69	<0.001	0.84	0.90

Data are presented means ± SD. *p* ≤ 0.05: significant. NA: not applicable. Significant differences between HIIT-CIT and HIIT-PLA groups at baseline using *t* test significant differences. * significant intra-group differences between pre and post intervention using paired *t* test. Time (HIIT intervention) and time × group effects using repeated-measure ANCOVA (adjusted by age). TUG = Timed up and Go; 6MWT = 6 Min Walking Test; ULMS = Upper Limb Muscle Strength; ULMSr = Upper Limb Muscle Strength/ body weight; LLMS = Lower limb Muscle Strength; LLMSr = Lower limb Muscle Strength/body weight; MQ = Muscle Quality

**Table 4 jcm-07-00561-t004:** Characteristics and differences in energy balance (potential confounder) at baseline and after 12 weeks of intervention between groups.

Variables	HIIT-CIT (*n* = 26)			HIIT-PLA (*n* = 30)			*p*-Value	
	Pre	Post	*p* Value	Pre	Post	*p* Value	Time Effect Adjusted for Age	Time × Group Effect Adjusted for Age
Energy balance								
Total Kcal intake (kcal/day)	1963 ± 310	1826 ± 540	0.36	2211 ± 1032	2055 ± 479	0.73	0.35	0.29
Proteins intake (g/day)	83.6 ± 20.7	87.3 ± 5.7	0.65	86.9 ± 30.5	79.6 ± 20.6	0.61	0.77	0.76
Carbohydrates intake (g/day)	252 ± 64	222 ± 76	0.24	267 ± 137	251 ± 52	0.79	0.25	0.22
Lipids intake (g/day)	69.6 ± 18.4	64.9 ± 26.3	0.28	90.2 ± 51.6	79.9 ± 23.7	0.65	0.24	0.26
Number of steps (*n*/day)	6639 ± 3448	6110 ± 3334	0.88	6228 ± 3217	5501 ± 3468	0.27	0.23	0.18

Data are presented as means ± SD. *p* ≤ 0.05 significant. NA: not applicable. Significant differences between HIIT-CIT and HIIT-PLA groups at baseline using *t* test significant differences. * significant intra-group differences between pre and post intervention using paired *t* test. Time (HIIT intervention) and time×group effects using repeated-measure ANCOVA (adjusted for age).

**Table 5 jcm-07-00561-t005:** Number of responders following the intervention.

Variables	HIIT-CIT Group (*n* = 26)		HIIT-PLA Group (*n* = 30)		*p*-Value
	Δ Changes (%)	Responders (%)	Δ Changes (%)	Responders (%)	
Body composition					
BMI	−0.6 ± 3.8	65.8	−1.3 ± 6.6	53	0.62
WC	−3.1 ± 2.3	92	−2.1 ± 2.8	83	0.15
Total LM	0.9 ± 3.1	53.8	1.3 ± 3.4	63.3	0.59
Total FM	−2.9 ± 6.0	73.3	−0.9 ± 5.12	50	0.17
Gynoid FM	−2.2 ± 7.4	53.8	−0.7 ± 7.7	56.7	0.46
Android FM	−2.6 ± 7.3	65.4	−0.6 ± 5.1	46.7	0.25
Legs FM	−2.9 ± 5.6	73.1	−1.5 ± 6.9	63.3	0.40
Functional & aerobic capacities					
TUGn (%)	−12.3 ± 5.8	96.2	−10.8 ± 10.9	90	0.53
TUGf (%)	−16.1 ± 9.0	100	−11.8 ± 7.8	93.3	0.04 *
6MWT (%)	14.5 ± 11.6	92.3	13.2 ± 13.3	93.3	0.70
Estimated Vo_2max_	10.2 ± 8.1	92.3	9.3 ± 9.3	93.3	0.69
Unipedal balance	109.1 ± 174.0	80.8	82.9 ± 96.1	86.7	0.48
Chair Stand test	−20.5 ± 8.5	100	−17.1 ± 8.8	96.6	0.15
Alternate step test	16.1 ± 7.2	100	17.4 ± 9.5	100	0.56
Muscle function					
ULMS	9.3 ± 10.8	82.3	3.3 ± 11.5	53.3	0.04 *
Relative ULMS	10.9 ± 11.6	80.8	5.3 ± 16.0	56.7	0.018 *
UMQ	10.4 ± 15.5	76.9	10.7 ± 37.3	60	0.98
LLMS	12.3 ± 21.5	80.8	3.3 ± 12.1	58.3	0.07 *
Relative LLMS	13.9 ± 22.3	80.8	5.6 ± 16.4	62.5	0.10
LMQ	10.8 ± 22.7	73.1	1.9 ± 12.6	54.2	0.17 *
Muscle Power	26.6 ± 30.7	80.8	25.7 ± 25.3	89.7	0.90

Data are presented as means ± SD. Delta change calculation (%): ((post-pre)/pre) × 100. *p* ≤ 0.05: significant using Man-Whitney tests. * change considered clinically significant. WC = waist circumference; BMI = Body Mass Index; LM = Lean Mass; FM = Fat Mass; TUG = Timed up and Go; 6MWT = 6 Min Walking Test; ULMS = Upper Limb Muscle Strength; UMQ = Upper Muscle Quality; LLMS = Lower Limb Muscle Strength; LMQ = Lower limb Muscle Quality.

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
