# Peer review of "Effect of High-Intensity Interval Training Combined with L-Citrulline Supplementation on Functional Capacities and Muscle Function in Dynapenic-Obese Older Adults"

_jcm, 2018, doi:10.3390/jcm7120561_

Reviewer 1 Report

This paper aims to determine whether citrulline supplementation can boost the effects of HIIT in dynapenic elderly subjects. The study focuses solely on the effects of the intervention on clinically relevant outcome measures associated with defining sarcopenia and sarcopenic obesity. In this regard, the study is largely well-designed, well-controlled and well-interpreted. A limitation is that the study provides no mechanistic insight into the potential mediators of the effect of citrulline. 

I have the following points for clarification: 

1)     There is no obvious advantage of shortening dynapenia to DY or obesity to O… it does, however, add to the confusion. There are also a very large number of acronyms used throughout the paper. 

2)     A major justification for use of Citrulline comes from studies showing that increasing arginine availability (via citrulline supplementation) in circumstances where arginine availability is inadequate (i.e. malnutrition). However, you make the point in the discussion that your subject pool is not malnourished. Do you have data from your dietary analysis to suggest that their arginine levels would be inadequate? Or are you proposing effects of citrulline beyond meeting essential nutrient requirements?

3)     Line 79. Not being done before is not a valid justification for a study. Please rephrase to provide a rationale for doing the study.

4)     Line 88. Where is the primary analysis published?

5)     How is it possible that body mass reduces by 3.1kg in the cit group, but BMI only changes from 30.5 to 30.4, whereas in the control group, body mass reduces by 1.1kg and BMI reduces by 0.5 kg/m2? This would imply changes in body height…. Or some subjects that started and didn’t finish.

6)     TUGf – between group difference adjusted for age. This seems hard to believe. The delta for the cit group is 1.2, while the delta for the control group is 0.9… and since age is 3 years higher in the control group it seems unlikely that adjusting for age would make this difference meaningfully significant. Of course for interventions the magnitude of the change is just as relevant as the statistical significance reached. With this in mind, effect sizes would likely give more useful information about the magnitude and likelihood of any beneficial effects due to Citrulline.  

 Line 279 – It is unclear what you mean by ‘significantly and clinically’. On a similar note, you have referred to some changes as ‘clinically significant’ but have failed to define what this is?

7)     Statistical analysis for table 3. If I have understood correctly, you have used a 2-way ANCOVA for analyzing differences in the response to HIIT between Cit and Pla groups. The within group comparison represents a change due to HIIT. However, it is not clear what the between group comparison represents. Based on table 3 and Figure 2, it appears as through the between group comparison compares both pre and post measures in Pla with pre and post measures in CIT… but I note you refer to this as an interaction in the legend. Please describe the statistical analysis in more detail.

8)     Age adjustment: an ANCOVA is only effective if the co-variate is a significant contributor to the variation in the outcome measure. In your data, is there a relationship between age and the various functional measures? If so, does it follow the expected relationship (i.e. functional parameters get worse with age)?

9)     Lines 316-319. What criteria do you use to define a ‘responder’? You would expect that 50% of subjects to perform better on a purely random basis. Furthermore, there seems to have been no familiarization performed for the functional tests…

10)  Lines 326-331 and table 6. These statements are not accurate. The Placebo group started the study with 22 of 30 (73%) classified as Type II dynapenic, whereas the Cit group started with only 16 of 26 (62%) classified as Type II dynapenic… In my opinion, these numbers are not close enough to say that they were ‘almost the same in both groups’… Especially when you try to make the point that the prevalence of type II dynapenia was higher in the Pla group (18 of 30, 60%) compared to the Cit group (12 of 26, 46%). In fact, both of these are non-significant differences and the reduction in % of Type II sarcopenia prevalence was 13% in the Pla and 16% in the CIT group which could hardly be considered different.

Author Response

COMMENTS AND SUGGESTIONS FOR AUTHORS

This paper aims to determine whether citrulline supplementation can boost the effects of HIIT in dynapenic elderly subjects. The study focuses solely on the effects of the intervention on clinically relevant outcome measures associated with defining sarcopenia and sarcopenic obesity. In this regard, the study is largely well-designed, well-controlled and well-interpreted. A limitation is that the study provides no mechanistic insight into the potential mediators of the effect of citrulline. 

Authors’ response:  We agree with the reviewer that mechanistic measurements would be of added value to our clinical relevant findings. However, we don’t have them available and it would take considerable amount of resources to address this point. Finally, and more importantly, we don’t have enough biological materials (number of tissue samples) to be able to answer adequately to this comment due to our design (secondary analysis). Thus, the mentioned limitation was added within the “limitations of the study” section (i.e.discussion) as « Finally, another limitation is that the study provides no mechanistic insight into the potential mediators (i.e mitochondrial activity, GH axis, nitric oxide (NO) etc…) of the citrulline effect.” (P20; lines 896-898).

In addition, as mentioned in answer #3-reviewer 1, we also added some information about potential mechanisms of action. 

 I have the following points for clarification: 

1) There is no obvious advantage of shortening dynapenia to DY or obesity to O… it does, however, add to the confusion. There are also a very large number of acronyms used throughout the paper. 

Authors’ response:We fully agree with the reviewer. The acronyms have been removed and the words are now written in full throughout the text. Please note that we don’t change the following acronyms HIIT, PLA and CIT because they appear very often in the text and are easy to understand.

 2) A major justification for use of Citrulline comes from studies showing that increasing arginine availability (via citrulline supplementation) in circumstances where arginine availability is inadequate (i.e. malnutrition). However, you make the point in the discussion that your subject pool is not malnourished. Do you have data from your dietary analysis to suggest that their arginine levels would be inadequate? Or are you proposing effects of citrulline beyond meeting essential nutrient requirements? 

Authors’ response:This is an important point. Thank you for pointing it out. The following paragraph has been added to the discussion section:

“We could hypothesize that arginine availability is inadequate not only in malnourished elderly but also in all older adults. Arginine levels in dietary intakes are not relevant since most of it would be metabolized or taken up both the gut and liver whereas CIT bypass these organs. As it has been shown, most of the CIT effect on explants of adipose tissue in rats were mediated through nitric oxide (NO) [21]. Thus, CIT could contribute to enhance nitric oxide effects in older adults [21].” (P19, lines 855-860)

 3) Line 79. Not being done before is not a valid justification for a study. Please rephrase to provide a rationale for doing the study.

Authors’ response:The rationale for the study has been rephrased according to the reviewer comment.

« Finally, nutrition in combination with exercise is considered optimal for maintaining muscle function. In this sense, Figueroa et al. showed that vibration/strength training combined with CIT induced greater improvement on leg fat-free mass in obese post-menopausal women [25]. However, no study has investigated the potential effects of CIT supplementation combined with an exercise intervention in older men and women and, especially, in those at risk of physical disability. Due to the potential benefits of CIT or HIIT, examine their combined effect in at risk population, such as dynapenic-obese older adults, is important. Therefore, the potential effects of CIT supplementation in older individuals with obesity and low muscle strength need to be studied to ascertain its benefits applications. » (P4, lines 121-129)

 4) Line 88. Where is the primary analysis published?

Authors’ response:The primary study is under process. In fact, from this larger and primary study we collected enough and sufficient number of biological tissue (muscle/ adipose tissue). In fact, it is well recognized that it takes considerable more resources and time to complete these type of analysis (muscle or adipose tissues) and to integrate these physiological findings.

We hope that this point won’t be a limitation to the publication of this interesting sub-study with clinical important results for the readership containing independent analysis. Finally, even if it is a sub-analysis by its design, we believe that this study could be considered as an independent research.

 5) How is it possible that body mass reduces by 3.1kg in the cit group, but BMI only changes from 30.5 to 30.4, whereas in the control group, body mass reduces by 1.1kg and BMI reduces by 0.5 kg/m2? This would imply changes in body height…. Or some subjects that started and didn’t finish.

Authors’ response: We would like to thank the reviewer to pointing out this difference between body weight and BMI. Based on his comment, we returned back to the data and observed a mistake in the reported body weight (in the PLA group only) and the correction has been done.  The following values are the good one and have been changed in the “results” section (P11, lines 391-394) and in Table 2 (p12): PRE : 82.6±12.5 kg and POST : 82.1±10.9 kg.  Thus, these values are now coherent with the changes of BMI.

 6) TUGf – between group difference adjusted for age. This seems hard to believe. The delta for the cit group is 1.2, while the delta for the control group is 0.9… and since age is 3 years higher in the control group it seems unlikely that adjusting for age would make this difference meaningfully significant. Of course for interventions the magnitude of the change is just as relevant as the statistical significance reached. With this in mind, effect sizes would likely give more useful information about the magnitude and likelihood of any beneficial effects due to Citrulline.  

Authors’ response:This is a relevant comment. Note that between group difference is still significant for TUGf with or without adjustment for age respectively) (p-values 0.03 and<0.001).  As suggested, the effect sizes have been also added in the text as follow (line 623-631 p15): “HIIT+CIIT group improved also more fast-paced “Timed Up and Go” test (-16.1 ± 8.99 vs. -11.8 ± 7.76; p=0.04; effect size: λ=0.012), absolute (9.33 ± 10.8 vs. 3.29 ± 11.5; p=0.04; effect size: λ=0.08) and relative (10.9 ± 11.6 vs. 5.26 ± 16.0; p=0.018; effect size: λ=0.08) upper limb muscle strength and tend to improve more lower limb muscle strength (12.3 ± 21.5 vs. 3.26 ± 12.1; p=0.07; effect size: λ=0.02) than HIIT+PLA (Table 5). In this sense, the percentage of responders is ranged between 53.8% (lean mass and gynoid fat masses) and 100% (fast-paced “Timed Up and go” and chair stand tests, alternate step test) in the HIIT + CIT group and between 46.7% (android fat mass) and 100% (alternate step test) in the HIIT + PLA group. Overall, the rate of responders is very high for all functional capacities (>80%) in both groups.”

 7) Line 279 – It is unclear what you mean by ‘significantly and clinically’. On a similar note, you have referred to some changes as ‘clinically significant’ but have failed to define what this is?

Authors’ response:We agree we the reviewer comments and we delete the words “clinically significant” from our article to be clearer. In fact, as mentioned and published previously, the TUG clinical cut point is +3 sec (Okumiya K, Matsubayashi K, Wada T, Kimura S, Doi Y, Ozawa T: Effects of exercise on neurobehavioral function in community-dwelling older people more than 75 years of age. J Am Geriatr Soc 1996, 44:569-72). Regarding handgrip strength, the cut-off is +6.5kg (Jae Kwang Kim, MD, PhD, Min Gyue Park, MD, and Sung Joon Shin, MD. What is the Minimum Clinically Important Difference in Grip Strength? Clin Orthop Relat Res. 2014 Aug; 472(8): 2536–2541).

 8) Statistical analysis for table 3. If I have understood correctly, you have used a 2-way ANCOVA for analyzing differences in the response to HIIT between Cit and Pla groups. The within group comparison represents a change due to HIIT. However, it is not clear what the between group comparison represents. Based on table 3 and Figure 2, it appears as through the between group comparison compares both pre and post measures in Pla with pre and post measures in CIT… but I note you refer to this as an interaction in the legend. Please describe the statistical analysis in more detail.

Authors’ response:We agree that the description of statistical analysis was confusing. The « statistical analysis » paragraph (P10, lines 346-356) and the legend below the tables (Tables 2, 3 and 4) and figure (figure 2) have now been improved. 

Statistical analysis: “Data distribution was tested with the Kolmogorov test. Quantitative variables were expressed by mean±standard deviation (SD). Qualitative variables were expressed in percentage. An independent sample t-test was used to identify between-group baseline differences  and the Chi-squared test or Fisher test to compare frequency of observations between groups or dynapenia prevalence.  A two-way repeated-measure analysis of covariance (ANCOVA adjusted for age) was used to estimate time (HIIT intervention) and time*group (PLA vs. CIT group) effects.  A responder is define as an individual who improve his physical parameters;  a non responder is inversely a person who do not change or decline his physical parameters.  All calculations were performed using SPSS 25.0 program (Chicago, IL, USA) and Statistica 10 software. p ≤ 0.05 was considered statistically significant.”

Legend: « Table 2 Legends: Data are presented as means ± SD. p≤0.05: significant. NA: non applicable. asignificant differences between HIIT-CIT and HIIT-PLA groups at baseline using a t test. significant differences * significant intra-group differences between pre and post intervention using paired t test. Time (HIIT intervention) and time*group effects using repeated-measure ANCOVA (adjusted by age). BW= Body Weight; BMI=Body Mass Index; WC= waist circumference; LM= Lean Mass; FM= Fat Mass.”

 9) Age adjustment: an ANCOVA is only effective if the co-variate is a significant contributor to the variation in the outcome measure. In your data, is there a relationship between age and the various functional measures? If so, does it follow the expected relationship (i.e. functional parameters get worse with age)? 

Authors’ response: It is well referenced in the scientific literature (see few examples below) that age is associated with a loss of muscle mass and a reduction in functional and motor abilities. It is therefore necessary to adjust our results on age. In addition, at baseline, age was significantly different in the 2 groups, which reinforces our choice to adjust the results for this confounding variable. Finally, we would like to inform the reviewer that with or without adjustment for age our results and conclusion are identical. A sentence in this sense has been added in result section (P11, lines 391-392).

Examples:

-      Cruz-jentoft et al. Age Ageing. 2010 Jul;39(4):412-23.

-      Bohannon RWet al. Physiotherapy. 2011 Sep;97(3):182-9

-      Himann J et al. Med Sci Sports Exerc. 1988;20(2):161–6

-      Studenski S et al.  JAMA. 2011;305(1):50–8.

-      L. Eduardo Cofré Lizamaet al., PLoS One. 2014; 9(10): e110757.

-      Shih-JungChenget al. International Journal of Gerontology214 ; 8 (4) : 197-202

 10)  Lines 316-319. What criteria do you use to define a ‘responder’? You would expect that 50% of subjects to perform better on a purely random basis. Furthermore, there seems to have been no familiarization performed for the functional tests… 

Authors’ response:we thank the reviewer to highlight our missing but important information about responder / non-responder definition. A responder is defined as an individual who improves his physical parameters whereas a non-responder is inversely a person who do not change or decline his physical parameters on a purely statistical basis. It does not take into account the clinical significance of the change.

This sentence has been added In statistical section (P10, lines 353-354).

 11)  Lines 326-331 and table 6. These statements are not accurate. The Placebo group started the study with 22 of 30 (73%) classified as Type II dynapenic, whereas the Cit group started with only 16 of 26 (62%) classified as Type II dynapenic… In my opinion, these numbers are not close enough to say that they were ‘almost the same in both groups’… Especially when you try to make the point that the prevalence of type II dynapenia was higher in the Pla group (18 of 30, 60%) compared to the Cit group (12 of 26, 46%). In fact, both of these are non-significant differences and the reduction in % of Type II sarcopenia prevalence was 13% in the Pla and 16% in the CIT group which could hardly be considered different.

Authors’ response:The paragraph has been reworded as follow, to provide some statistical information obtained using fisher or Chi-sqaure tests and a simple description of table 6 (without interpretation) (P16, lines 718-725).

“Finally, the prevalence of type I and II dynapenia at baseline was 27% and 73% in the HIIT+PLA group vs. 38% and 62% in the HIIT+CIT group. At the end of the intervention, the prevalence of type II dynapenia was 60% in the HIIT+PLA group compared to 46% in the HIIT+CIT group. No difference at baseline (p= 0.399) and post intervention (p= 0.116) between both groups was observed. More interestingly, 12% of the subjects became non dynapenic in the HIIT+CIT group after the intervention period whereas they were dynapenic at the beginning of the study. These results are very important for physicians and clinicians aiming to counteract or prevent dynapenia”

Reviewer 2 Report

The manuscript (number: jcm-393180), titled: “Effect of high-intensity interval training combined with L-citrulline supplementation on functional capacities and muscle function in dynapenic-obese older adults” appears very interesting.

Sarcopenia has been recognized as a disease entity recognised by an ICD-10-CM (M62.84) code and this should lead to an increase in availability of diagnostic tools and treatment strategies. The scientific community has been working hard in finding algorithms to screen, assess and diagnose sarcopenia (Cruz-Jentoft et al. 2018). However, uncertainties on muscle mass quantification are still existing, especially in obese subjects. I think the choice to evaluate the effect of HITT and CIT supplementation on dynapenia - instead of sarcopenia - in obese subjects is interesting and appropriate.

The study appears well approached, several aspects have been considered by authors. However, some limitations are present. Specific comments are the following:

1.      I suggest to add the method used for cognitive impairment assessment in the section “Material and Methods”. I understand that the manuscript is a secondary analysis of a study previously reported, however reader may be interested in details of test used.    

2.      I have some doubts on TUG execution and cut-off values indicated by authors in the text. Usually, in Time Up and Go test the subject rises from an arm chair, walks 3 meters, turns, walks back, and sits down again. A duration time above 13.5 seconds indicates an increased risk of falling [Steffen et al., Phys Ther 2002;82(2):128-137], and 12 seconds or less has been suggested as screening tool for mobility status [Bischoff HA et al., Age and Ageing2003;32:315–320].

Authors considered a 4-meters distance (even if references were related to TUG-3-meters walking distance). Can authors mention a reference for method they used and cut-off values they considered?

Author Response

COMMENTS AND SUGGESTIONS FOR AUTHORS

The manuscript (number: jcm-393180), titled: “Effect of high-intensity interval training combined with L-citrulline supplementation on functional capacities and muscle function in dynapenic-obese older adults” appears very interesting. 

Sarcopenia has been recognized as a disease entity recognised by an ICD-10-CM (M62.84) code and this should lead to an increase in availability of diagnostic tools and treatment strategies. The scientific community has been working hard in finding algorithms to screen, assess and diagnose sarcopenia (Cruz-Jentoft et al. 2018). However, uncertainties on muscle mass quantification are still existing, especially in obese subjects. I think the choice to evaluate the effect of HITT and CIT supplementation on dynapenia - instead of sarcopenia - in obese subjects is interesting and appropriate.

The study appears well approached, several aspects have been considered by authors. However, some limitations are present. Specific comments are the following:

 1) I suggest to add the method used for cognitive impairment assessment in the section “Material and Methods”. I understand that the manuscript is a secondary analysis of a study previously reported, however reader may be interested in details of test used.   

Authors’ response:Thank you for this comment. The method used to assess cognitive status (The Montreal Cognitive Asssessment : MoCA) has been described in the « material and methods » section (P7, lines 222-226) as follow: « The validated Montreal Cognitive Assessment (MoCA) was used to assess cognitive status. The MoCA test evaluates visuospatial abilities, executive functions, orientation, language, attention, concentration, and working memory. An extra point to the total score is given if the subject has ≤12 years of education. A cut-off value of ≥26 out of 30 points was established to differ healthy subjects from mild cognitive impairment subject [30].”

 2) I have some doubts on TUG execution and cut-off values indicated by authors in the text. Usually, in Time Up and Go test the subject rises from an arm chair, walks 3 meters, turns, walks back, and sits down again. A duration time above 13.5 seconds indicates an increased risk of falling [Steffen et al., Phys Ther 2002;82(2):128-137], and 12 seconds or less has been suggested as screening tool for mobility status [Bischoff HA et al., Age and Ageing2003;32:315–320]. Authors considered a 4-meters distance (even if references were related to TUG-3-meters walking distance). Can authors mention a reference for method they used and cut-off values they considered?

Authors’ response: Itis an error, thank you to note it. It has been corrected in the text. In our lab we are generally doing 4 meter walking distance and a 3 m Timed Up and Go test. In this study, we used the latter.

« Walking speed was estimated using the « Timed Up & Go » test (in sec). This test consists to a complete task of standing from a chair, walking a 3-meter distance and sitting down again [32]was performed in comfortable and self-paced  and in fast-paced  walking speed. A duration above 30 seconds indicates limited mobility and an increased risk of falling whereas a duration of less than 20 seconds indicates appropriate mobility with the subject being likely to be independent in activities of daily living [33] ». 
